# IMAGPose: A Unified Conditional Framework for Pose-Guided Person Generation

**Fei Shen,  Jinhui Tang**[*]
Nanjing University of Science and Technology
{feishen, jinhuitang}@njust.edu.cn

## Abstract

Diffusion models represent a promising avenue for image generation, having demonstrated competitive performance in pose-guided person image generation. However, existing methods are limited to generating target images from a source image and a target pose, overlooking two critical user scenarios: generating multiple target images with different poses simultaneously and generating target images from multi-view source images. To overcome these limitations, we propose IMAG-Pose, a unified conditional framework for pose-guided image generation, which incorporates three pivotal modules: a feature-level conditioning (FLC) module, an image-level conditioning (ILC) module, and a cross-view attention (CVA) module. Firstly, the FLC module combines the low-level texture feature from the VAE encoder with the high-level semantic feature from the image encoder, addressing the issue of missing detail information due to the absence of a dedicated person image feature extractor. Then, the ILC module achieves an alignment of images and poses to adapt to flexible and diverse user scenarios by injecting a variable number of source image conditions and introducing a masking strategy. Finally, the CVA module introduces decomposing global and local cross-attention, ensuring local fidelity and global consistency of the person image when multiple source image prompts. The three modules of IMAGPose work together to unify the task of person image generation under various user scenarios. Extensive experiment results demonstrate the consistency and photorealism of our proposed IMAG-Pose under challenging user scenarios. The code and model will be available at https://github.com/muzishen/IMAGPose.

## 1   Introduction

Pose-guided person generation [51, 29] aims to transform a source person image into a target person image under a specific pose while maintaining appearance consistency. It has many applications, including virtual reality, film production, and e-commerce. Besides, the generated images can be used to enhance the performance of downstream tasks, such as person re-identification [32, 54, 48, 34].

Early methods [4, 57, 42] usually are developed based on generative adversarial networks (GANs). However, GAN-based methods easily suffer from the instability of the min-max training objective and the difficulty of generating high-quality images in a single forward pass. Recently, methods based on diffusion models [1, 22, 11, 36] are becoming increasingly popular in the community. As a promising alternative to GAN for image generation, diffusion models utilize the source image and target pose as conditions. They generate the target image through a multi-step denoising process instead of completing it in a single step. So, diffusion models help better retain the input information. For example, PIDM [1] proposes a texture diffusion module that inserts features of the source image, extracted by a frozen encoder, into different stages of the diffusion

---

[*]Corresponding author

38th Conference on Neural Information Processing Systems (NeurIPS 2024).

model's UNet. Similarly, PoCoLD [11] further constrains the correspondence between the person and the pose by introducing additional 3D Densepose [19] annotations and interacting with the appearance features of the source image extracted by the frozen encoder. Additionally, the PCDMs [36] introduce a three-stage diffusion model that is used for predicting the global features of the target image, generating a coarse-grained person image, and refining texture details.

Despite the above methods based on diffusion models' impressive results compared to GAN-based approaches, existing methods still exhibit 2 shortcomings. (1) **For methods based on diffusion models, they neglect underlying texture detail information.** As these methods employ a frozen image encoder, e.g., CLIP, to extract appearance features from the source image, which is trained on a general image dataset, it is not a specialized feature extractor for person images and can only extract high-level semantic features. (2) **For existing methods, as shown in Figure 1, they overlook two critical user scenarios: generating multiple target images with different poses simultaneously and generating target images from multi-view source images.** In practice, users often

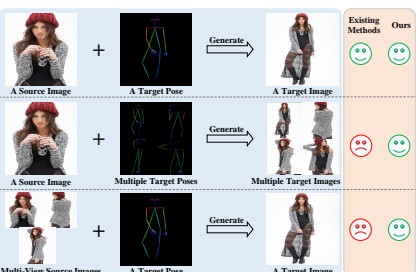

Figure 1: Existing methods can only support generating a target image from one source image and one target pose.

need to generate multiple consistent target images based on different poses simultaneously, such as in portrait photography and clothing displays. In addition, users often upload multiple source images to provide rich details for generating a target image. However, the fixed architecture of existing methods limits the ability of users to make arbitrary modifications and extensions.

We propose a unified conditional framework named IMAGPose to address the above issues. For the first issue, we devise that the feature-level conditioning (FLC) module extracts low-level texture features from the VAE encoder and high-level semantic features from the image encoder. Because the frozen VAE can almost reconstruct images losslessly, it can easily extract low-level information without training. By leveraging the VAE's encoding capability, we fuse the source image's low-level and high-level information, addressing the absence of a dedicated person image feature extractor. To address the second issue, we propose the image-level conditioning (ILC) module randomly injecting various numbers of source image conditions and introducing a masking strategy to adapt to different user scenarios. Specifically, we combine all target images into a joint target, and then the masking strategy randomly masks the target image into a joint mask according to user scenarios. Meanwhile, the ILC extracts the pose from the source image and combines it with the target pose to form a joint pose. Therefore, IMAGPose can inject image-level source image conditions from the input end, achieving a one-to-one correspondence between the person's image and pose. Further, we present the cross-view attention (CVA) module, introducing decomposable global and local cross-attention to ensure the person images' local fidelity and global consistency when any user scenarios. We summarize the contributions of this paper as follows,

- We are the first to explore pose-guided image generation tasks under different conditions and propose a unified conditional framework for synthesizing high-fidelity person images.

- We present the FLC module, which addresses the issue of missing texture features due to the absence of a dedicated person image feature extractor.

- We devise the ILC module to inject a variable number of source image conditions and introduce a masking strategy, achieving an alignment of images and poses.

- We develop a CVA module to decompose global and local cross-attention, ensuring the joint image's local fidelity and global consistency when multiple-view source image prompts.

- Extensive experiments on two challenging datasets show that proposed IMAGPose surpasses existing models. We also conduct a user study and applications for downstream tasks to comprehensively evaluate the advantages of IMAGPose.

## 2   Related Work

**Based on GAN Methods.** Early approaches [24, 25, 37, 57, 37, 49, 30] view the synthesis task as conditional image generation, utilizing conditional generative adversarial networks (CGANs) [26] to generate target images based on source appearance images and target poses. For example, PG$^2$ [24]

proposed a two-stage method that refines the generated images in an adversarial manner. To decouple pose and appearance information, VUNet [8] suggests learning pose-independent features to tackle the complex structure problem of pose space transformation. Then, ADGAN [25] introduced a character attribute decomposition module to explicitly parse the human body and blend and insert conditions. VariGANs [53] combine variational inference with GANs to generate multi-view images from a single image, refining results from coarse to fine [5, 7, 6]. However, GAN-based methods [24, 25, 37] often suffer from issues such as blurred texture details and instability of the min-max training objective. To address the issue of blurred texture details, Grigorev et al. [9] proposed a CNN-based framework that first performs pose warping, followed by texture repair. Unselfie [18] introduced a pipeline that first identifies the target's neutral pose, then repairs appearance textures, and finally perfects and synthesizes the character in the background. However, these multi-stage methods require the introduction of multiple additional models and costs, impacting the efficiency of generation.

**Based on Diffusion Models.** Diffusion models have found extensive applications in virtual dressing [33], story generation [35], and portrait animation [45]. To address the instability of GAN's min-max training, methods based on diffusion models [13, 44, 40, 38, 31, 28] are gradually becoming more popular in the community. The core idea is to start from a simple noise vector and gradually transform it into a high-quality image through multiple denoising iterations, making it suitable for high-fidelity and context-aware generation. Besides unconditional generation [14, 39, 41], various methods [50, 47, 27] have been introduced to incorporate user-provided control signals into the generation process, thereby achieving more controllable image generation. For example, PIDM [1] proposed a texture diffusion module that uses the source image features extracted by the frozen encoder as conditions to be inserted into different stages of UNet. CFLG [22] proposed a mixed-granularity attention module to inject multi-scale fine-grained character appearance features into the generation process. PoCoLD [11] further constrains the correspondence between person images and poses by introducing additional 3D Densepose [19] annotations of poses and source image appearance features extracted by the frozen encoder. Subsequently, PCDMs [36] proposed a three-stage diffusion model that predicts global features under the target pose, aligns images and poses, and then refines texture details. Besides, APS [16] progressively couples target pose and appearance for effective human image synthesis. PoSynDA [20] simulates 3D pose distributions to address the lack of 2D-3D correspondences, enhancing data diversity. MotionEditor [43] uses a dual-branch structure that decouples key-value queries, preserving the background and character appearance for efficient content editing. However, the frozen encoders introduced in these diffusion model-based methods are trained on generic image datasets and can only extract high-level semantic features of person, easily overlooking low-level texture details. Furthermore, in real scenarios, these models do not support generating multiple target images at once and target generation under multi-view image prompts.

## 3 Method

Figure 2 illustrates IMAGPose, a unified conditional framework that encompasses 3 core modules: feature-level conditioning (FLC) module, image-level conditioning (ILC) module, and cross-view attention (CVA) module. This framework aims to generate high-fidelity and high-quality target images in any user scenario. The FLC module combines low-level texture features from the VAE encoder with high-level semantic features from the image encoder, which addresses the issue of losing texture details due to lacking a dedicated person image feature extractor. (Section 3.2). The ILC module introduces a masking strategy for the target image and incorporates a variable number of source image conditions. This module aligns the image and pose, making it suitable for different user scenarios. (Section 3.3). The CVA module proposes the decomposition of global and local cross-attention. This ensures the entire person images' local fidelity and global consistency.(Section 3.4).

### 3.1 Preliminaries

Diffusion models belong to a class of generative models. And these models are trained to reverse a diffusion process, which systematically introduces Gaussian noise to the data through a fixed Markov chain over a series of timesteps, denoted as $t$. Concurrently, a denoising model is trained to generate samples starting from this Gaussian noise. The training objective of a diffusion model, denoted as $\epsilon_\theta$ and parameterized by $\theta$, typically employs a mean square error loss $L_{DM}$. For each timestep $t$, this loss is defined as follows,

$$L_{DM} = \mathbb{E}_{x_0,\epsilon,c,t}\|\epsilon - \epsilon_\theta(x_t, c, t)\|^2,  \tag{1}$$

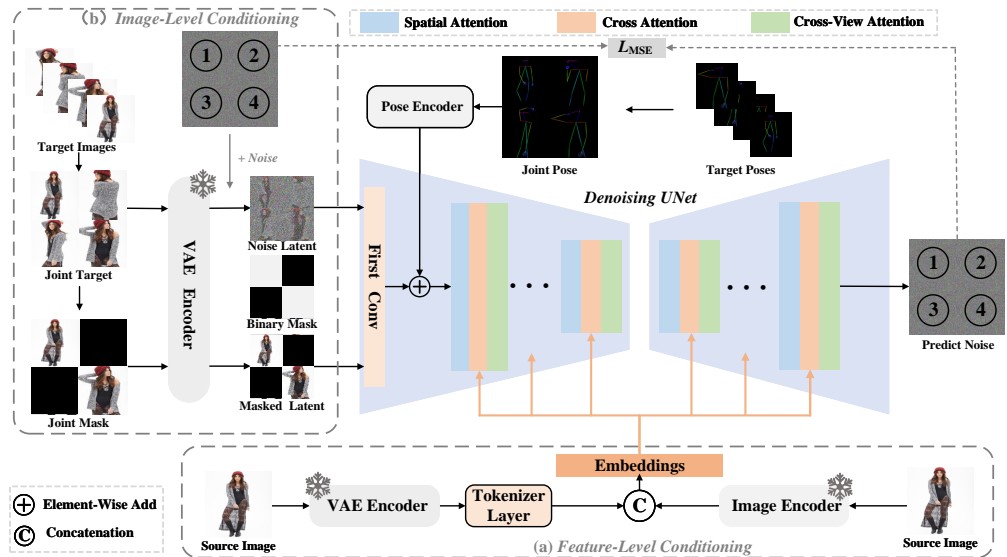

Figure 2: The IMAGPose is a unified conditional framework designed to generate high-fidelity and high-quality target person images under various conditions. IMAGPose aims to address the issue of detail texture loss, achieve an alignment of person images and poses, and ensure the person images' local fidelity and global consistency.

here, $x_0$ represents the original data, supplemented by a condition $c$. The timestep $t$ of the diffusion process is represented by $t \in [0, T]$. The noisy data at the $t$ step, denoted as $x_t$, is defined as $\alpha_t x_0 + \sigma_t \epsilon$. In this context, $\alpha_t$ and $\sigma_t$ are predefined functions of $t$ that dictate the diffusion process.

Once the model $\epsilon_\theta$ is trained, images can be synthesized from random noise through an iterative process. During the sampling stage, the predicted noise is calculated based on the predictions of both the conditional model $\epsilon_\theta(x_t, c, t)$ and the unconditional model $\epsilon_\theta(x_t, t)$, using classifier-free guidance [15] according to Eq. 2.

$$\hat{\epsilon}_\theta(x_t, c, t) = w\epsilon_\theta(x_t, c, t) + (1 - w)\epsilon_\theta(x_t, t), \tag{2}$$

where $w$ is the guidance scale used to adjust the condition $c$.

## 3.2 Feature-Level Conditioning Module

Existing methods typically depend on frozen encoders to extract appearance features from source images. However, these frozen encoders are trained on general image datasets, not explicitly designed for person image feature extraction. So, it can only capture high-level semantic features of the source image. To address this, as depicted in Figure 2(a), given that the frozen VAE can almost reconstruct images losslessly without training, we propose the FLC module, which simultaneously extracts low-level texture information from a VAE encoder [17] and high-level semantic information from an image encoder. Specifically, the FLC module consists of a frozen VAE encoder, an image encoder, and a trainable tokenizer layer. Assume that the source image $x \in \mathbb{R}^{c \times h \times w}$, where $c$, $h$, and $w$ represent the channel, height, and width of the image, respectively. First, we extract the latent features $F_l \in \mathbb{R}^{4 \times h/8 \times w/8}$ of the source image from the frozen VAE encoder while simultaneously extracting the high-level semantic features $F_H \in \mathbb{R}^{n \times d}$ of the person image from the frozen image encoder, where $n$ and $d$ denote the token length and each token dimension, respectively. Then, we convert latent features $F_l$ into texture features $F_L \in \mathbb{R}^{m \times d}$ through a learnable tokenizer layer consisting of a 2D convolution and a flattened operation, where $m$ stand for the length of $F_L$ token. We ensure that the dimension of each token from the texture features $F_L$ matches the dimension of each token from the semantic features $F_H$ through the kernel size of the 2D convolution. Finally, according to Eq. 3, we concatenate $F_L$ and $F_H$ along the token length to obtain the person's appearance features.

$$F_C = concat(F_L, F_H), \tag{3}$$

where $F_C \in \mathbb{R}^{(m+n) \times d}$. Therefore, FLC leverages the frozen VAE encoder's ability to preserve detailed information, addressing the issue of not having a dedicated person image feature extractor.

### 3.3 Image-Level Conditioning Module

As previously mentioned, existing methods only support the generation setting of one source image and one target pose, which limits user expectations in other use scenarios. To address this, we propose the ILC module to inject a random variable number of source image conditions and introduce a masking strategy to adapt to different user scenarios. As shown in Figure 3(a), when generating three target images based on three target poses, the ILC first masks the three target images to be generated, then combines them with one source image to form a joint mask, while extracting the target pose from the source image and combining it with the three given target poses to form a joint pose; As shown in Figure 3(b), when generating a target image based on three source images, the ILC first masks one target image and then

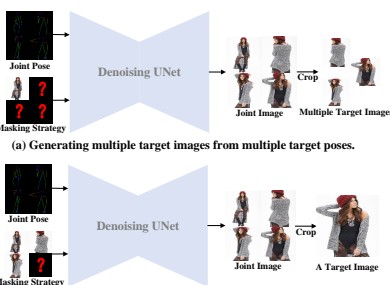

(a) Generating multiple target images from multiple target poses.

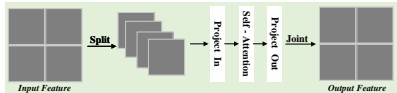

(b) Generating a target image from multi-view source image prompts.

Figure 3: The masking strategy flexibly unify different user scenarios.

combines it with three source images to form a joint mask while extracting the poses from the three source images and combining them with one given target pose to form a joint pose. This achieves a one-to-one correspondence between the person's image and pose.

Specifically, we use a frozen VAE to extract latent space features from the joint target and add noise to obtain the noise latent. Similarly, we use a frozen VAE to get the masked latent from the joint mask. The ILC module concatenates the noise latent and masked latent along the channel dimension. Then, we introduce a binary mask, the width and height of which are the same as the noise latent, with 0 and 1 representing the masked and unmasked parts, respectively. This ensures that the model correctly distinguishes the areas of the target image to be generated. Therefore, the input channels of our model are 9, with the channel numbers of noise latent and masked latent both being 4, and the binary mask being 1. Besides, noted that existing methods ignore the image-level conditions of the source image and only inject the feature-level conditions of the source image through the cross-attention module. In contrast, as shown in Figure 2(b), we inject the image-level conditions of the source image combined with the masking strategy, achieving richer inherent context information and spatial structure alignment between the person's image and pose.

### 3.4 Cross-View Attention Module

To ensure that each target image in the generated joint objectives possesses realistic visual content and maintains consistency in the person subject images, inspired by [2, 10], we propose a cross-view attention (CVA) module to decompose global and local cross-attention. As shown in Figure 2, the

Figure 4: Illustration of the CVA module.

CVA module is embedded after each cross-attention module in the denoising UNet. The schematic diagram of the CVA module is shown in Figure 4, which mainly consists of the split, projection layer, self-attention layer, and joint. We first take the output of the feature by cross-attention as the input features of CVA, split the global input features into four smaller local person features, and then add a new temporal dimension to learn the attention between each local character image. Subsequently, the four person features are reshaped back into one global output feature, ignoring the temporal dimension. This approach allows IMAGPose to capture both global and local attention, providing a more nuanced understanding of the relationships between different parts of the persons' image.

For the pose condition, we introduced a pose encoder identical to ControlNet [2] for injection after the first convolutional layer. Therefore, the loss function $L_{\mathrm{MSE}}$ of IMAGPose according to Eq. 4, as follows. Here, $F_C$, $F_I$ and $F_P$ denote the feature of the FLC module, the feature of the ILC module and pose feature, respectively.

$$L_{\mathrm{MSE}} = \mathbb{E}_{x_0,\epsilon,F_C,F_I,F_P,t}\|\epsilon - \epsilon_\theta\big(x_t, F_C, F_I, F_P, t\big)\|^2. \qquad (4)$$

In the inference stage, we use classifier-free guidance according to Eq. 5.

$$\hat{\epsilon}_\theta(x_t, F_C, F_I, F_P, t) = w\epsilon_\theta(x_t, F_C, F_I, t) + (1 - w)\epsilon_\theta(x_t, F_P, t). \qquad (5)$$

---

[2]https://github.com/lllyasviel/ControlNet

Table 1: Quantitative comparison of the proposed IMAGPose with several state-of-the-art models.

| Dataset | Methods | SSIM ($\uparrow$) | LPIPS ($\downarrow$) | FID ($\downarrow$) |
|---|---|---|---|---|
| DeepFashion [21] (256 × 176) | Def-GAN [37] | 0.6786 | 0.2330 | 18.457 |
| | PATN [57] | 0.6709 | 0.2562 | 20.751 |
| | ADGAN [25] | 0.6721 | 0.2283 | 14.458 |
| | PISE [49] | 0.6629 | 0.2059 | 13.610 |
| | GFLA [30] | 0.7074 | 0.2341 | 10.573 |
| | DPTN [51] | 0.7112 | 0.1931 | 11.387 |
| | NTED [29] | 0.7182 | 0.1752 | 8.6838 |
| | CASD [56] | 0.7248 | 0.1936 | 11.373 |
| | PoCoLD [11] | 0.7310 | 0.1642 | 8.0667 |
| | PIDM [1] | 0.7312 | 0.1678 | 6.3671 |
| | CFLD [22] | 0.7378 | 0.1519 | 6.8040 |
| | PCDMs [36] | 0.7444 | 0.1365 | 7.4734 |
| | **IMAGPose (Ours)** | **0.7561** | **0.1284** | **5.8738** |
| DeepFashion [21] (512 × 352) | CocosNet2 [55] | 0.7236 | 0.2265 | 13.325 |
| | NTED [29] | 0.7376 | 0.1980 | 7.7821 |
| | PIDM [1] | 0.7419 | 0.1768 | 5.8365 |
| | PoCoLD [11] | 0.7430 | 0.1920 | 8.4163 |
| | CFLD [22] | 0.7478 | 0.1819 | 7.1490 |
| | PCDMs [36] | 0.7601 | 0.1475 | 7.5519 |
| | **IMAGPose (Ours)** | **0.7718** | **0.1396** | **5.6298** |
| Market-1501 [54] (128 × 64) | Def-GAN [37] | 0.2683 | 0.2994 | 25.364 |
| | PTN [57] | 0.2821 | 0.3196 | 22.657 |
| | GFLA [30] | 0.2883 | 0.2817 | 19.751 |
| | DPTN [51] | 0.2854 | 0.2711 | 18.995 |
| | PIDM [1] | 0.3054 | 0.2415 | 14.451 |
| | PCDMs [36] | 0.3169 | 0.2238 | 13.897 |
| | **IMAGPose (Ours)** | **0.3282** | **0.2104** | **12.659** |

## 4 Experiments

**Datasets.** We conducted experiments on the DeepFashion dataset [21], which consists of 52,712 high-resolution images of fashion models, and the Market-1501 dataset [54], which includes 32,668 low-resolution images with diverse backgrounds, viewpoints, and lighting conditions. We extracted the skeletons using OpenPose [3] and followed the dataset splits provided by [1]. It's important to note that the person IDs of the training and testing sets do not overlap for both datasets.

**Metrics.** We conducted a comprehensive evaluation of the model, including both objective and subjective metrics. Objective metrics include the structural similarity index measure (SSIM) [46], learned perceptual image patch similarity (LPIPS) [52], and fréchet inception distance (FID) [12]. On the other hand, subjective assessments focus on user-oriented metrics, such as the percentage of real images misclassified as generated (R2G) [24], the percentage of generated images misclassified as real (G2R) [24], and the percentage of images considered superior among all models (Jab) [37].

**Implementations.** We conduct experiments on 8 NVIDIA V100 GPUs. Our configuration can be summarized as follows: (a) We use the pre-trained Stable Diffusion V1.5 [3] and modified the first convolutional layer to accommodate additional conditions. Unless otherwise specified, we use Dinov2-G/14 [4] as the image encoder. In the tokenizer layer, both the kernel size and stride of the 2D convolution are 16, and the dimensions of the input and output channels are 4 and 768, respectively. (b) Following [1, 36], we train our model on the DeepFashion dataset with sizes of 256 × 176 and 512 × 352. For the Market-1501 dataset, we used images of size 128 × 64. (c) In the masking strategy, we defaulted to randomly occluding 1-4 images. (d) The model is trained for 300k steps using the AdamW optimizer with a learning rate of $5e^{-5}$. Each batch size is 4, and a linear noise schedule of 1000 time steps is applied. (e) In the inference stage, we used a DDIM sampler with 20 steps, and set $w$ to 2.0 in the guidance scale.

### 4.1 Main Results

IMAGPose supports generating one or multiple target images under different conditions based on user scenarios. For a fair comparison and adaptation to the IMAGPose framework, we use three

---

[3] https://huggingface.co/runwayml/stable-diffusion-v1-5

[4] https://github.com/facebookresearch/dinov2

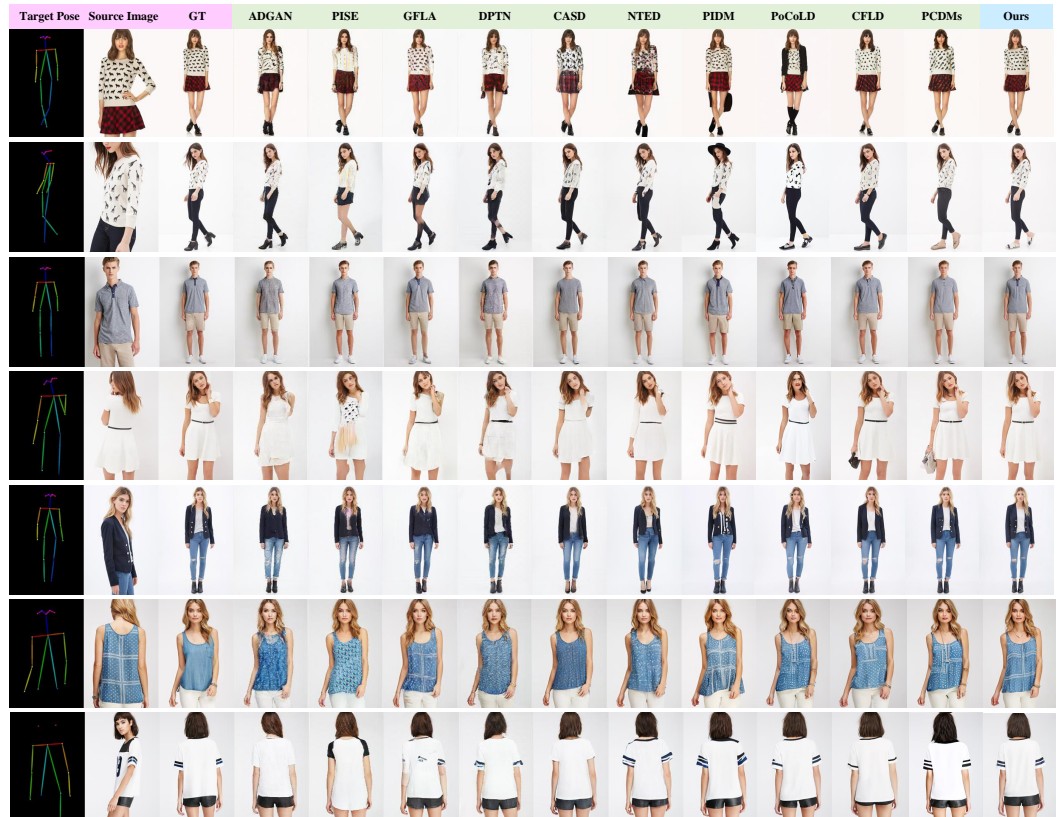

Figure 5: Qualitative comparisons with several state-of-the-art models on the DeepFashion dataset.

black images for padding in the masking strategy and replicate the target pose 3 times in the joint pose during the inference stage. Then IMAGPose generates 3 same target images, and we randomly select one generated image for qualitative and quantitative evaluation. Unless otherwise specified, the above settings are the default configurations.

**Quantitative Results.** From Table 1, we quantitatively compared proposed IMAGPose with several state-of-the-art methods including Def-GAN [37], PATN [57], ADGAN [25], PISE [49], GFLA [30], DPTN [51], NTED [29], CocosNet2 [55], CASD [56], PoCoLD [11], PIDM [1], PCDMs [36] and CFLD [22]. Compared to other generative methods, based on diffusion models [11, 1, 36, 22] perform significantly better, especially proposed IMAGPose, which outperforms all methods. For example, IMAGPose significantly leads the GAN-based method ADGAN on the SSIM metric since the GAN framework's unstable training fails to generate high-quality images. Compared to CFLD, which is also based on a diffusion model framework, IMAGPose demonstrates absolute performance advantages at both $256 \times 176$ and $512 \times 352$ scales. Because existing architectures overlook texture details and only inject appearance features using a generic image encoder.

Table 1 also shows the quantitative results on the Market-1501. Like the DeepFashion, IMAGPose outperforms all SOTA methods in SSIM, LPIPS, and FID. Compared to NTED, which uses fine-grained texture features, ours shows more significant advantages, thanks to VAE's inherent ability to reconstruct low-level texture. Despite PCDMs' additional refinement model, IMAGPose provides richer context information by injecting source image features at both image and feature levels.

**Qualitative Results.** Figure 5 visually compares our IMAGPose with other SOTA methods on the DeepFashion dataset. The results lead to several conclusions: (1) As shown in the first to third rows, diffusion-based methods capture minute clothing textures, but IMAGPose generates higher-quality details than other diffusion models. (2) In cases of large-scale pose transformations (as shown in the fourth and fifth rows), CFLD and PCDMs exhibit clear overfitting to the pose (i.e., objects do not should appear on the hand ), only our method can generate images aligned with the target and consistent in detail texture. (3) As shown in the sixth to seventh rows, when generating complex textures and invisible areas, IMAGPose reasonably infers detailed textures, providing better visual consistency even if the results are inconsistent with the target. In summary, the proposed method consistently produces more realistic and lifelike person images, demonstrating the advantages of the image-level and feature-level conditions proposed by IMAGPose. Please see C.1 for more examples.

**User Study.** The above quantitative and qualitative comparisons reflect the significant advantages of IMAGPose in generating results. Further, we conducted a user study with 50 volunteers. This study includes comparisons with basic facts (i.e., R2G and G2R) and other methods (i.e., J2b). The higher the score for these three indicators, the better the performance. As shown in Figure 6, IMAGPose exhibits commendable performance on all three indicators on the DeepFashion dataset. For instance, volunteers judged our generated images as 58.5% (G2R) real

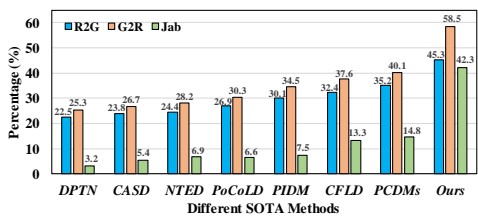

Figure 6: User study results on DeepFashion in terms of R2G, G2R and Jab metric. Higher values in these three metrics indicate better performance.

images, nearly 18.4% higher than the second-best model. The proposed IMAGPose scored 42.3% on the Jab index, indicating our method is more popular. Please see C.3 of the Appendix for more detail.

**Uniformity.** IMAGPose only needs to be trained once to support multiple user scenarios. Here, IMAGPose simulates two user scenarios: generating a target image given a source image and a target pose, and generating a target image given multiple source images and a target pose. As shown in Figure 7, both T1 and T2 provide one source image and one target pose, similar to the previous SOTA. The difference is that **T1** replicates the target pose in the joint pose, while **T2** replicates the source image in the joint image. **T3** randomly uses multiple different source images.

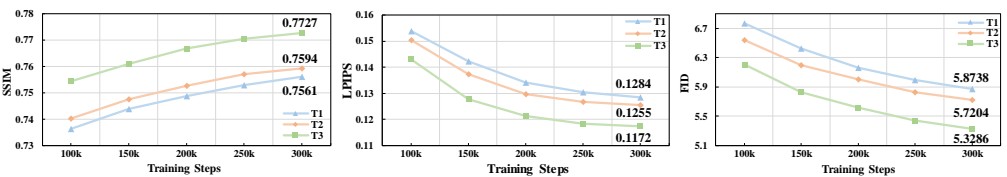

Figure 7: Quantitative comparison of IMAGPose under different user scenarios on the DeepFashion dataset.

We found that under these 3 different settings, compared to Table 1, T1, T2, and T3 demonstrate strong competitiveness on all metrics, indicating that IMAGPose can adapt to the above scenarios only once training. For example, T2 surpasses all previous SOTA methods and beats T1, suggesting the ILC module can effectively capture the appearance features of a person by replicating the same source image multiple times during the inference stage. Then, when using multiple different source images to generate a target image, T3 achieves 0.7727, 0.1172, and 5.3286 on SSIM, LPIPS, and FID, respectively, outperformers T2 and other SOTA methods, indicating that IMAGPose has uniformity when provided multiple different source images.

Furthermore, IMAGPose can also adapt to the other scenario: generating multiple target images from a source image and multiple target poses, i.e., IMAGPose* in Figure 8. Compared with the original IMAGPose, IMAGPose* slightly reduces the SSIM performance but still surpasses other diffusion model-based methods, which proves that even when inferring multiple different target images at once, IMAGPose* can still maintain high quality and fidelity. Regarding speed, due to the architectural

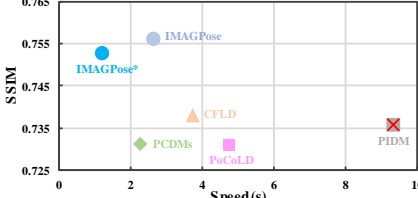

Figure 8: Results of speed and performance.

advantage of the masking strategy in ILC, it is nearly 8 times faster than PIDM, and nearly 3 times faster than PoCoLD and CFLD. Noted that both the default settings of IMAGPose and IMAGPose* are highly competitive regarding generation quality and speed. Therefore, once our method is trained, users can choose the appropriate inference settings according to their needs in different scenarios. The visual results of IMAGPose in different user scenarios are shown in Figure 9, which is consistent with our above analysis. Overall, IMAGPose can unify different tasks in real-world scenarios. For more results and discussions, please refer to C.2 of the Appendix.

### 4.2 Ablations

**Effectiveness of the Module.** Table 2 demonstrates the impact of different modules on the results of IMAGPose. **B0** denotes IMAGPose removes the CVA module, the texture features from the VAE in the FLC module, and the image-level conditions in the ILC module, which results in a model with only 5 input channels. **B1** represents, based on B0, the addition of the CVA module. **B2** denotes, based on B0, adding the VAE's texture detail features, resulting in a complete FLC module. **B3** indicates,

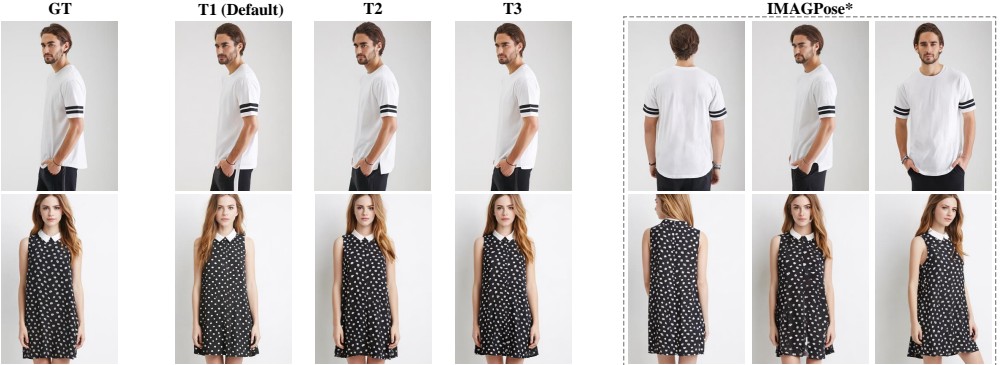

Figure 9: Visual comparison of our model's uniformity across different user scenarios.

based on B1, that adding image-level conditions forms a complete ILC module with 9 input channels. **B4** represents IMAGPose using both the FLC and ILC modules, while ignoring the CVA module.

As shown in Table 2, IMAGPose outperforms other settings on three metrics and gradually improves performance as more modules are included. Specifically, B1 surpasses B0 on SSIM, LPIPS, and FID, indicating that the CVA module can enhance local fidelity and global consistency. B2 and B3 outperform B1 by 0.0125 and 0.0172 on SSIM, suggesting that FLC and ILC can effectively provide exemplary texture detail guidance and ensure alignment of images and

Table 2: Ablation study on DeepFashion.

| Methods | SSIM (↑) | LPIPS (↓) | FID (↓) |
|---------|----------|-----------|---------|
| B0 | 0.7235 | 0.1704 | 8.2153 |
| B1 | 0.7281 | 0.1646 | 7.8209 |
| B2 | 0.7406 | 0.1471 | 7.3284 |
| B3 | 0.7453 | 0.1390 | 6.3655 |
| B4 | 0.7528 | 0.1326 | 6.0242 |
| **Ours** | **0.7561** | **0.1284** | **5.8738** |

poses. Moreover, noted that the combination of the three modules brings additional performance improvements. For example, our method outperforms B2, B3, and B4 on all metrics, respectively.

**Model Variant.** Table 3 summarizes the variants based on IMAGPose. We keep other settings unchanged and only modify the base model and image encoder. The overall performance differences between various versions of the base diffusion model and image encoder are insignificant, indicating that the base diffusion model and image encoder have a minimal impact on the results of the generated images. Specifically, the results suggest that when 'Base.' is SD V1.5, Dinov2-G/14 slightly outperforms on all metrics. Furthermore, when using the same image encoder, SD V2.1 is superior to SD V1.5 in most cases.

Table 3: Performance of variant version.

| Base. | Image Enc. | SSIM (↑) | LPIPS (↓) | FID (↓) |
|-------|-----------|----------|-----------|---------|
| SD V1.5 | CLIP-B/14 | 0.7516 | 0.1364 | 6.1342 |
| | CLIP-bigG/14 | 0.7548 | 0.1331 | 5.9645 |
| | CLIP-H/14 | 0.7552 | 0.1296 | 6.0231 |
| | Dinov2-B/14 | 0.7541 | 0.1343 | 5.9286 |
| | Dinov2-L/14 | 0.7556 | 0.1323 | 5.9432 |
| | Dinov2-G/14 | **0.7561** | **0.1284** | 5.8738 |
| SD V2.1 | CLIP-B/14 | 0.7562 | 0.1350 | 6.2736 |
| | CLIP-bigG/14 | 0.7535 | 0.1281 | 6.2239 |
| | CLIP-H/14 | 0.7542 | 0.1266 | 6.1239 |
| | Dinov2-B/14 | 0.7553 | 0.1304 | 5.9120 |
| | Dinov2-L/14 | **0.7579** | 0.1276 | 6.1315 |
| | Dinov2-G/14 | 0.7572 | 0.1294 | **5.8543** |

### 4.3 Application

Table 4 evaluates the applicability of images generated by IMAGPose in downstream tasks, i.e., person re-identification. Initially, we randomly selected 20%, 40%, 60%, and 80% subsets from the training set of the Market-1501 dataset, ensuring at least one image per identity, thereby creating a new dataset. We adopt BoT [23] as the base network and perform baseline training on each data subset for a fair comparison. Then, we incorporate images generated by IMAGPose. These generated images are synthesized from randomly selected images with the same identity and pose from the new dataset. The Rank1 results are shown in Table 4. The experimental results indicate

Table 4: Comparison with SOTA on person re-identification. *Standard* denotes not using generated person images.

| Methods | Percentage of real images | | | |
|---------|------|------|------|------|
| | 20% | 40% | 60% | 80% |
| *Standard* | 33.4 | 56.6 | 64.9 | 69.2 |
| PTN [57] | 55.6 | 57.3 | 67.1 | 72.5 |
| GFLA [30] | 57.3 | 59.7 | 67.6 | 73.2 |
| DPTN [51] | 58.1 | 62.6 | 69.0 | 74.2 |
| PIDM [1] | 61.3 | 64.8 | 71.6 | 75.3 |
| PCDMs [36] | 63.8 | 67.1 | 73.3 | 76.4 |
| **IMAGPose** | **66.4** | **69.2** | **75.1** | **77.6** |

that the IMAGPose algorithm significantly improves re-identification performance compared to the *Standard*. Moreover, IMAGPose consistently demonstrates superior performance in re-identification tasks compared to SOTA methods.

## 5 Conclusion

This paper presents IMAGPose, a unified conditional framework for pose-guided image generation. This framework addresses the limitations of existing methods by enabling the generation of multiple target images with different poses simultaneously and from multi-view source images. IMAGPose

incorporates three pivotal modules: a Feature-Level Conditioning (FLC) module, an Image-Level Conditioning (ILC) module, and a Cross-View Attention (CVA) module. The FLC module effectively combines low-level texture features with high-level semantic features, addressing the issue of missing detail information. The ILC module aligns images and poses to adapt to diverse user scenarios, while the CVA module ensures local fidelity and global consistency when multiple source image prompts. The proposed framework has demonstrated competitive performance in both qualitative and quantitative evaluations, showing consistency and photorealism under challenging user scenarios. This work represents a significant step forward in the field of image generation, opening up new possibilities for user interaction and customization. **Limitations.** (1) IMAGPose supports generating images in different user scenarios and significantly improves the synthesis quality. However, due to computational limitations, IMAGPose currently supports generating up to four images at a time. Future work will continue to explore how to generate more target images in one forward process. (2) The performance of the model is subpar for generating cartoon characters and non-photorealistic styles because our training data consists of photorealistic human images for fair comparison. In future work, we plan to include a broader range of data and design style transformation modules to overcome this bias.

## Acknowledgments and Disclosure of Funding

This work is supported by the National Natural Science Foundation of China (Grant No. 61925204, and 62332010). The authors would thank all Reviewers and ACs of OpenReview for the thorough review and insightful suggestions.

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

## Supplementary Material

This supplementary material offers a more detailed exploration of the experiments and methodologies proposed in the main paper. Section A provides a series of symbols and definitions for enhanced comprehension. Section B delves deeper into the implementation specifics of our experiments. Section C presents additional experimental outcomes, including a broader range of qualitative comparison examples with state-of-the-art methods, more results on different user scenarios, and a detailed explanation of our user studies. Section D discusses potential societal harms and impacts.

## A    Some Notations and Definitions

The notations and definitions used in this paper are shown in Table 5.

Table 5: Some notations and definitions.

| Notation | Definition |
|---|---|
| $x_0$ | Target image |
| $\epsilon$ | Gaussian noise |
| $c$ | Additional condition |
| $t$ | Timestep |
| $\theta$ | Diffusion model |
| $w$ | Guidance scale |
| $x_t$ | Noisy data at $t$ step |
| $F_H$ | High-Level semantic feature of source image |
| $F_L$ | Low-Level texture feature of source image |
| $F_C$ | Feature of FLC module |
| $F_I$ | Feature of ILC module |
| $F_P$ | Feature of joint point |

## B    Implement Details

Table 6: Hyperparameters for the IMAGPose.

| Hyperparameters | IMAGPose |
|---|---|
| Diffusion Steps | 1000 |
| Noise Schedule | Linear |
| Optimizer | AdamW |
| Weight Decay | 0.01 |
| Batch Size | 4 |
| Iterations | 300k |
| Learning Rate | $5e-5$ |
| Base Model | Stable Diffusion V1.5 |
| Image Encoder | Dinov2-G/14 |

As shown in Table 6, our experiments are conducted on 8 NVIDIA V100 GPUs. We use the pre-trained Stable Diffusion V1.5 and modify the first convolutional layer to accommodate additional conditions. Dinov2-G/14 is used as the image encoder. In the tokenizer layer, we set both the kernel size and stride of the 2D convolution to 16, and the dimensions of the input and output channels are set to 4 and 768, respectively. We introduce a pose encoder identical to ControlNet, injected after the first convolutional layer, to handle the pose condition. Our model is trained on the DeepFashion dataset, following previous studies, with image sizes of 256x176 and 512x352. For the Market-1501 dataset, we use images of size 128x64. As part of our masking strategy, we default to randomly occluding 1-4 images. The model is trained for 300k steps using the AdamW optimizer with a learning rate 5e-5. We use a batch size of 4 and apply a linear noise schedule of 1000 time steps. During the inference stage, we use a DDIM sampler with 20 steps and set the guidance scale to 2.0.

# C   Additional Results

## C.1   More Qualitative Comparisons for IMAGPose

We provide additional examples for comparison with the state-of-the-art (SOTA) methods in Figure 10.

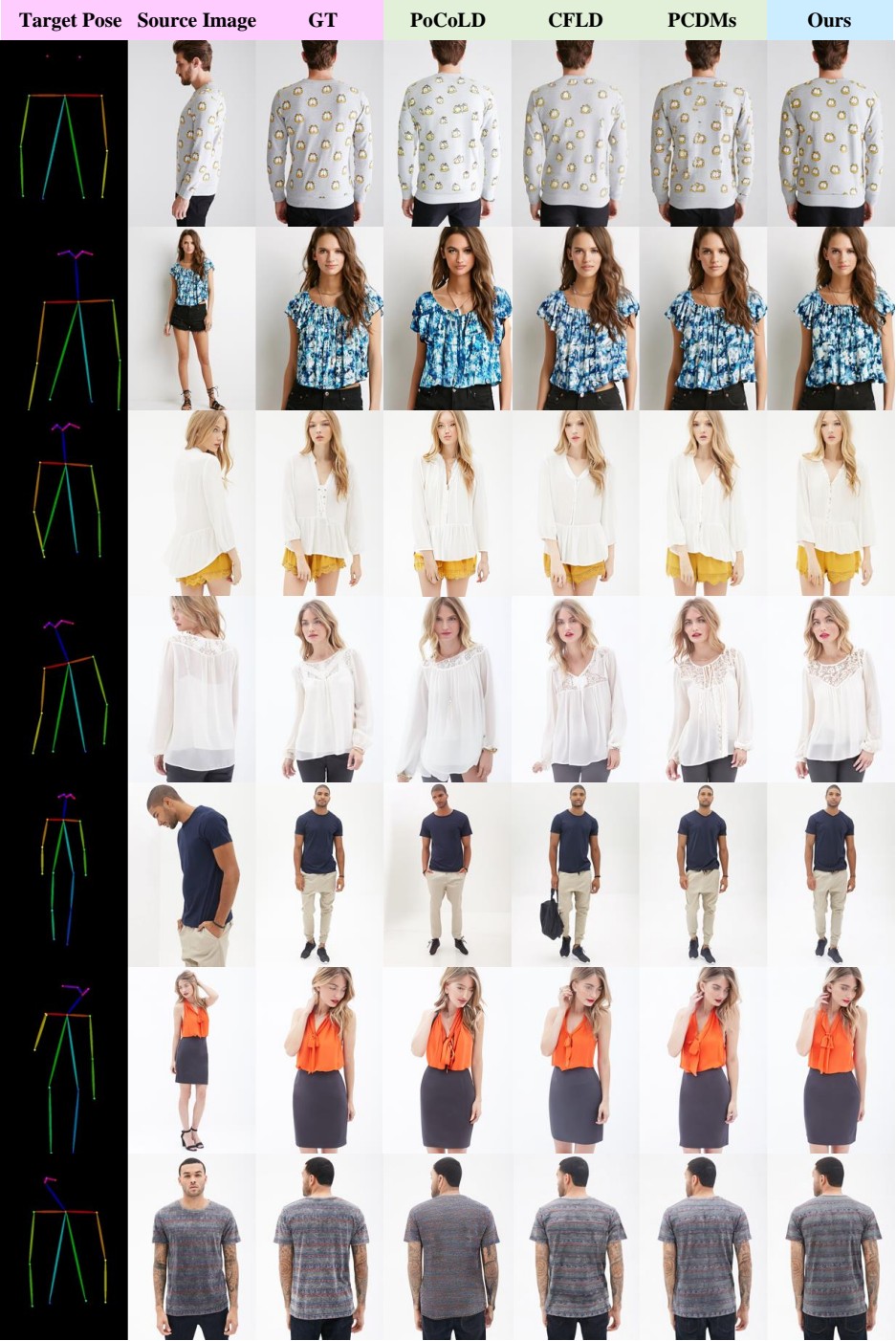

Figure 10: More qualitative comparisons between IMAGPose and SOTA methods on the DeepFashion dataset.

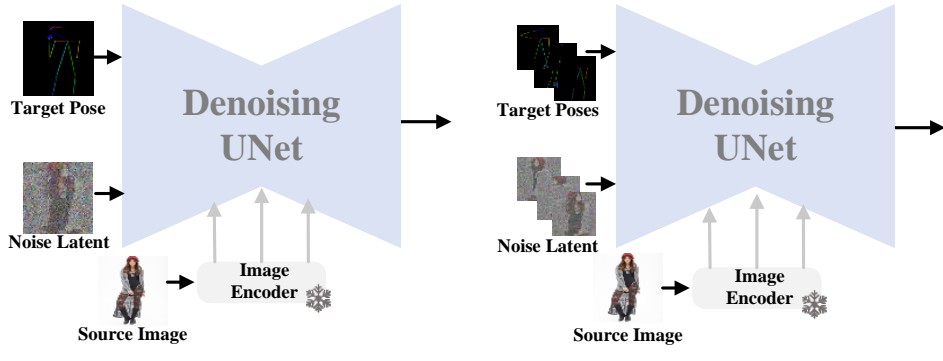

(a) Common Framework : Generating a target image     (b) M1 : Generating multiple target images
from a single source image and a single target pose.    with different poses simultaneously.

Figure 11: (a) The schematic diagram of the common frameworks based on existing diffusion models can only support generating a target image from a single source image and a single target pose. (b) During the development of IMAGPose, we devised a proprietary model to address the scenarios of generating multiple target images with different poses simultaneously.

Table 7: Comparison results between different architectures on DeepFashion.

| Methods | SSIM ($\uparrow$) | LPIPS ($\downarrow$) | FID ($\downarrow$) |
|---------|------|-------|-----|
| PIDM [1] | 0.7312 | 0.1678 | 6.3671 |
| M1 | 0.7145 | 0.1743 | 6.9406 |
| **Ours** | **0.7561** | **0.1284** | **5.8738** |

## C.2 More Discussion on Different User Scenarios

As discussed in the main text, previous pose-guided image generation methods only support scenarios where a target image is generated from a single source image and a target pose, as shown in Figure 11 (a). However, in practical user scenarios, it is also common and important to generate multiple target images with different poses, as shown in Figure 1. To address these scenarios, we develop a proprietary models M1 to generate multiple target images with different poses simultaneously during the development of IMAGPose, as shown in Figure 11(b). Similar to [2, 10], M1 concatenates different noise latents along the batch dimension. Then, we chose the PIDM [1] based on the diffusion model as the representative for generating a target image from a single source image and a target pose. M1 model are also included in our additional comparisons, as shown in Table 7.

The results in Table 7 indicate that while we can address the generation of target images from multiple target poses by training a dedicated model M1, this approach increases the training burden and introduces complexity for users. Users must continually switch between models to meet their specific needs when generating target images. Moreover, from a performance metric perspective, the dedicated model M1 does not offer any advantages, as it only injects semantic information from the source image using a frozen image encoder. Most importantly, none of the current frameworks support the task of generating target images through multiple image prompts. In contrast, our proposed ILC module introduces a masking strategy to unify target image generation under different user scenarios. As shown in Figure 2 (b), we also incorporate the masking strategy to inject image-level conditions from the source image, and enhance the texture features of the VAE to achieve richer context information and alignment between the image and pose.

As previously mentioned, once IMAGPose is trained, it can accommodate different user scenarios. For generating a target image from a single source image and a target pose, IMAGPose only needs to replicate the target pose multiple times in the joint pose during inference. For generating multiple target images from a single source image and multiple target poses at once, IMAGPose only needs to combine different poses in the joint pose during inference. For generating a target image from multiple source images and a single target pose, IMAGPose only needs to combine different source images in the joint image during inference. Therefore, IMAGPose has the ability to unify pose-guided image generation under different user scenarios.

### C.3 User Study

To validate the disparity between the generated images and the real ones, and to evaluate the superiority of our method compared to existing technologies, we conducted a user study involving 50 volunteers. (1) For the R2G and G2R metrics, we selected 40 test samples at random and used each model to generate a corresponding set of images. The volunteers were tasked with distinguishing between the 40 generated images and 40 real images. (2) For the Jab metric, we randomly selected 40 pairs of source images and target poses, and each model was tested to generate a corresponding set of images. The users were then asked to select the highest quality and most faithful image from the side-by-side views. The user study allowed us to evaluate which model produced the most realistic results based on human perception. Example questions are illustrated in  13.

## D   Ethics Statement

In this paper, we propose IMAGPose, a unified conditional framework designed to cater to various user scenarios for generating different human images. However, it is well-known that virtually all methods of human image synthesis, including ours, can be misused by malicious actors to create false content and disseminate misinformation. This is a significant concern that we fully acknowledge. Nevertheless, substantial progress has been made in the field of detecting and preventing such malicious tampering. Our work aims to provide valuable support for research and external audits in this area, helping to strike a balance between the value of our technology and the risks associated with unrestricted access. This ensures that our technology can be used safely and beneficially.

**T1 (Default)**  **T2**  **T3**  **IMAGPose***

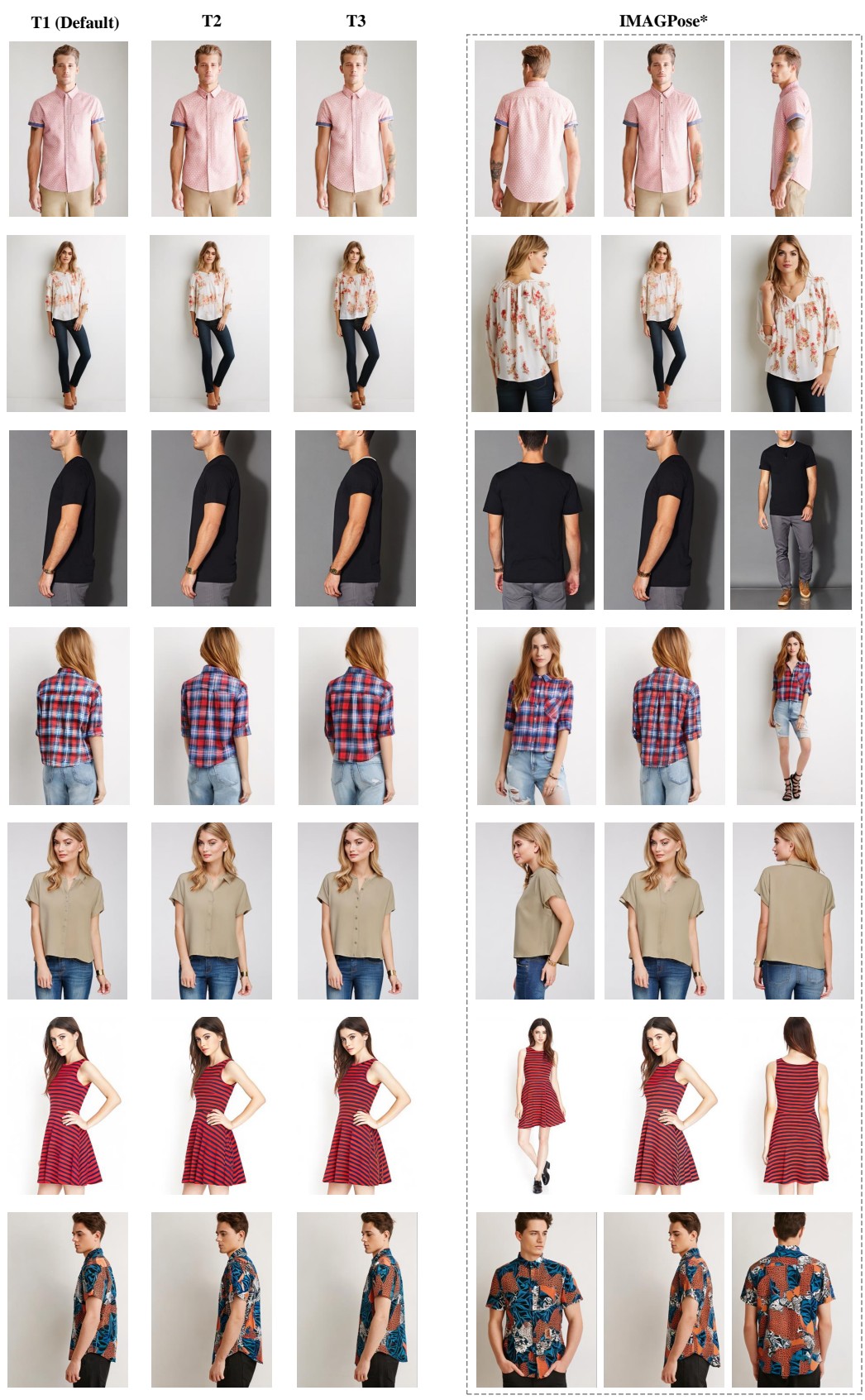

Figure 12: More visual comparison of our model's uniformity across different user scenarios.

23. **Please indicate whether the following image is real or fake.**

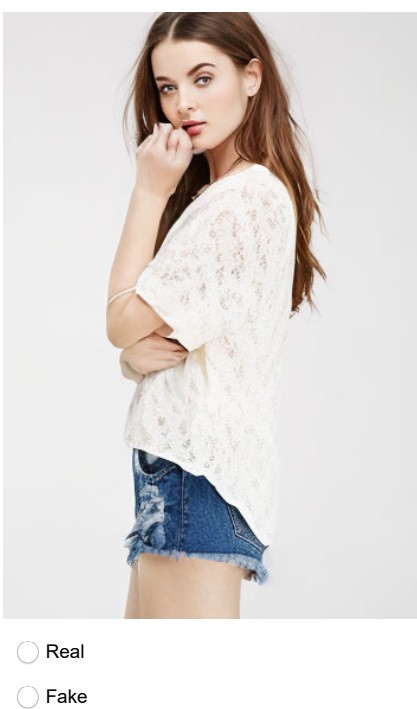

○ Real

○ Fake

Figure 13: An example question used in our user study for pose-guided person image synthesis.

