# OpenReview forum: "IMAGPose: A Unified Conditional Framework for Pose-Guided Person Generation"
_NeurIPS.cc/2024/Conference — NeurIPS 2024 poster_

### Official Review · Reviewer_Cm6N · 2024-06-29

**Soundness:** 2
**Presentation:** 1
**Contribution:** 2
**Rating:** 4
**Confidence:** 4

**Summary:**

This paper proposes a diffusion-based pose-guided image generation method. Specifically, given an image and a sequence of poses, this paper aims to generate images that follow the input poses and maintain the appearance of the input image. To capture the texture details of the input image, they propose to combine features from VAE and a pre-trained image encoder. They design an Image-level conditioning module, which combines four images into a joint representation.  The images that need to be predicted are masked. By doing this, their method can generate 1-3 images in one forward step, or use 1-3 condition images in the same time.

**Strengths:**

1. They propose to combine VAE and image encoder features as the condition, which helps the model to generate more details images.
2. Their Image-Level Conditioning design enables their models to condition on the dynamic number of input images.

**Weaknesses:**

1. Limited novelty. 1） The main contribution of this paper is forming four image latents into a 4-grid join latent, which is very similar to [1]. 2） The design of Sec 3.2 combines the VAE feature with the feature from Dino, which is similar to Animate Anyone[2], which uses CLIP image features as the condition.
2. Lack of thorough comparison methods. Given an RGB person image $\alpha$ and a pose image $\beta$, this paper aims to generate a person that follows the pose image  $\beta$ with the appearance of the $\alpha$, which can also be called pose transfer. Thus, this paper should also compare their results with pose-guided human pose transfer methods: Animate Anyone [2]. DreamPose [3], Follow your pose [4] …. Besides, this paper injects pose guidance in ControlNet way, they should also compare their performance with ControlNet.
3. Incorrect statements. In line 44-50, they claim that former diffusion-based generation using frozen image encoder pre-trained on general image dataset (not specific for person image), so these methods cannot extract texture detail information. However, Animate Anyone [2] trains a reference appearance net on large-scale human datasets, which can effectively encode detailed texture features.

[1] Kara, Ozgur, et al. "Rave: Randomized noise shuffling for fast and consistent video editing with diffusion models." Proceedings of the IEEE/CVF Conference on Computer Vision and Pattern Recognition. 2024.
[2] Hu, Li, et al. "Animate anyone: Consistent and controllable image-to-video synthesis for character animation." arXiv preprint arXiv:2311.17117 (2023).
[3] Karras, Johanna, et al. "Dreampose: Fashion image-to-video synthesis via stable diffusion." 2023 IEEE/CVF International Conference on Computer Vision (ICCV). IEEE, 2023.
[4] Ma, Yue, et al. "Follow your pose: Pose-guided text-to-video generation using pose-free videos." Proceedings of the AAAI Conference on Artificial Intelligence. Vol. 38. No. 5. 2024.

**Questions:**

1. Since this paper forms four images into a joint feature map, the self-attention in SD can conduct feature transformation among each images. Why do you split the joint feature map and conduct temporal attention among them (Sec 3.4)?
2. This paper evaluates their method solely on person images with clean background. How does this method perform on images with complex, meaningful backgrounds.

**Limitations:**

Yes, they have provided discussions about limitations and negative social impact.

---

> ### Author Rebuttal · Authors · 2024-08-07
>
> Dear Reviewer Cm6N
>
> We thank the reviewer for the positive feedback and valuable comments.
>
> **Q1: Limited novelty. 1） The main contribution of this paper is forming four image latents into a 4-grid join latent, which is very similar to [1]. 2） The design of Sec 3.2 combines the VAE feature with the feature from Dino, which is similar to Animate Anyone[2], which uses CLIP image features as the condition.**
>
> **Response:** (1) We sincerely disagree with your viewpoint. First, our method is not merely a simple 4-grid join latent. Our primary innovation involves first merging all target images into a unified target, followed by a carefully designed masking strategy that randomly masks these images into a joint mask, an approach not mentioned in reference [1]. Furthermore, our proposed ILC module aims to adapt to various user scenarios by randomly injecting different source image conditions and incorporating source image-level conditions, which are entirely different from the video editing tasks, motivations, and methods described in reference [1]. More importantly, IMAGPose introduces two new task settings and attempts to unify them within a single framework, achieving competitive results in each setting.
> (2) Please refer to the **shared response** on "Differences with technologies like Animate Anyone." We have added and discussed these differences.
>
> **Q2: Lack of thorough comparison methods. Given an RGB person image  and a pose image , this paper aims to generate a person that follows the pose image   with the appearance of the ,
> which can also be called pose transfer. Thus, this paper should also
> compare their results with pose-guided human pose transfer methods:
> Animate Anyone [2]. DreamPose [3], Follow your pose [4] …. Besides, this
> paper injects pose guidance in ControlNet way, they should also compare
> their performance with ControlNet.**
>
> **Response:** Thank you for your kind reminder. Following your suggestion,  as shown in **Figure 2 of the submitted PDF file**, we have added comparisons with Animate Anyone [2], DreamPose [3],  and ControlNet [5] on deepfashion dataset. It is important to note that we cannot directly compare our method with Follow Your Pose [4], as the input conditions for Follow Your Pose [4] are text and pose, and it does not support guided by image references. Quantitative and qualitative results demonstrate that IMAGPose achieves highly competitive performance. This is because references [2,3,5] focus on continuous pose-guided generation; our approach is specifically designed for scenarios involving non-continuous poses. Moreover, we wish to highlight that IMAGPose explores and unifies image generation across different user scenarios, including Scenario (1), generating one target image from one source image and one target pose; Scenario (2), generating one target image from multiple source images and one target pose; and Scenario (3), generating multiple target images from one source image and multiple target poses. However, references [2,3,5] are optimized and designed for a single scenario task, lacking the capability for unification and multitasking. For example, references [2-3] only support Scenario (3), and ControlNet [5] only supports Scenario (1). Neither accommodates the user's need to generate from multiple source images.
>
> |Methods| SSIM (↑)| LPIPS (↓) | FID (↓)|
> |----------|----------|----------|----------|
> | ControlNet| 0.6725| 0.2443 |15.8762|
> |DreamPose | 0.7161| 0.1694 | 8.2510|
> | Animate Anyone| 0.7386 | 0.1319 | 6.8024|
> |**Ours**|**0.7561** | **0.1284** | **5.8738**|
>
> **Q3: Incorrect statements. In line 44-50, they claim that former diffusion-based generation using frozen image encoder pre-trained on general image dataset (not specific for person image), so these methods cannot extract texture detail information. However, Animate Anyone [2] trains a reference appearance net on large-scale human datasets, which can effectively encode detailed texture features.**
>
> **Response:** We wholeheartedly agree with your perspective. As you mentioned, enhancing the model's ability to encode texture features [2] involves integrating an additional reference network and a large dataset. In other words, without these additions, it becomes relatively challenging. Following your suggestion, we have added a constraint requiring no additional modules and datasets. Thank you once again for your meticulous guidance.
>
> **Q4: Since this paper forms four images into a joint feature map, the self-attention in SD can conduct feature transformation among each images. Why do you split the joint feature map and conduct temporal attention among them (Sec 3.4)?**
>
> **Response:** Thank you for your insightful question. We split the joint feature map and apply temporal attention for the following reasons: (1) Splitting allows for a more detailed exploration of the temporal context of each image, helping to capture subtle changes. (2) This approach enhances the model’s ability to track and integrate dynamics across varying poses and scenes, which is crucial for accurately modeling scenarios involving extensive pose transitions or scene changes. (3) Joint features can mask or dilute individual image characteristics. Splitting the feature map allows for more targeted feature transformations, maintaining the integrity of each individual image.
>
> **Q5: This paper evaluates their method solely on person images with clean
> background. How does this method perform on images with complex,
> meaningful backgrounds.**
>
> **Response:** Please refer to the **shared response** regarding the "Results on Out-of-Domain Datasets." We have included additional visualization results on more diverse datasets.

---

> ### Author Response · Authors · 2024-08-11
> **Seeking Further Feedback**
>
> Dear Reviewer Cm6N:
>
> Again, thank you very much for the detailed comments.
>
> In our earlier response and revised manuscript, we have conducted additional experiments and provided detailed clarifications based on your questions and concerns.  As we are ending the stage of the author-reviewer discussion soon, we kindly ask you to review our revised paper and our response and consider adjusting the scores if our response has addressed all your concerns.  Otherwise, please let us know if there are any other questions.  We would be more than happy to answer any further questions.
>
> Best,
>
> Authors

---

> > ### Comment · Area_Chair_vE5F · 2024-08-12
> > **Let's engage in reviewer-author discussion**
> >
> > Dear Reviewer Cm6N,
> >
> > We look forward to seeing your comments on the authors' rebuttal and any further clarifications you may need.
> >
> > Thanks

---

> ### Comment · Reviewer_Cm6N · 2024-08-13
> **I’m inclined to change my final review to Borderline Reject.**
>
> Thank you for your detailed response.
>
> They claim to differ from [1] in several aspects. I appreciate their clarification, explaining that they can mask 1-3 images in the 4-grid setting, allowing for image generation beyond the two images used in [1]. However, it would be relatively straightforward to retrain [1] to ‘adapt to various user scenarios’ by simply modifying its masking strategy. Besides, I recommend the authors add some visualization results about how the self-attention in SD performs under the different masking strategies, \eg, the attention maps.
>
> Additionally, the four-grid setting reduces the resolution of each sub-image, as the generated sub-images are one-fourth the size of the original SD. Given that their focus is on image generation, which demands detailed, high-quality results, the output may not meet the desired level of quality.
>
> Finally, this paper presents a fairly complete work. Although the novelty of each module is relatively limited, when combined, it forms a cohesive and comprehensive study. Their response addresses some of my concerns, I am willing to change my final review to borderline reject. However, I still have reservations about the novelty of the work.

---

> ### Author Response · Authors · 2024-08-13
> **Thank you and expect more disucssions !**
>
> Dear Reviewer Cm6N,
>
> We sincerely thank you once again for your detailed suggestions and encouragement, such as “**a fairly complete work**” and “**a cohesive and comprehensive study.**” These comments have significantly enhanced our work and inspired us to pursue further research!
>
> **Q1: it would be relatively straightforward to retrain [1] to ‘adapt to various user scenarios’ by simply modifying its masking strategy.**
>
> **Response:** we respectfully disagree. We believe this overlooks a key contribution of our work: proposing two additional extended tasks from real-world user scenarios. Existing methods typically require training three separate models for generating images in these three different scenarios. In contrast, our proposed IMAGPose, through the design of the FLC, ILC, and CVA modules, achieves unification and delivers competitive performance across all scenarios. Additionally, we want to emphasize that our approach **is not simply about modifying the masking strategy.** We introduced an additional set of combined images as a condition (i.e., concatenating four images along the height and width) and then masking certain sub-images. This differs from [1], which directly uses latent space representations. By using image-level conditions, we provide more contextual information.
>
> **Q2: Attention maps.**
>
> **Response:** Due to NeurIPS policies, we are unable to provide additional experimental results (including anonymous links) at this stage. However, we can describe the attention visualization: ‘split image’ pays more attention to texture details compared to ‘joint image,’ as the latter focuses on global consistency while the former emphasizes detail consistency between frames. We will include this visualization and analysis in the final version of the paper.
>
> **Q3: Additionally, the four-grid setting reduces the resolution of each sub-image, as the generated sub-images are one-fourth the size of the original SD. Given that their focus is on image generation, which demands detailed, high-quality results, the output may not meet the desired level of quality.**
>
> **Response:**  we are concerned there might be a misunderstanding about the four-grid setting. In fact, our training and inference data are doubled in both height and width, meaning the output results  (every sub-image) are the same size as the original SD outputs. Additionally, extensive qualitative, quantitative, and user studies have demonstrated the strong competitive performance of our method.
>
> **Q4: I still have reservations about the novelty of the work.**
>
> **Response: ** We are concerned there may be a misunderstanding regarding the novelty of our work. **(1) This paper is the first to define three different pose-guided image generation tasks based on real-world user scenarios. (2) IMAGPose is the first to attempt solving these three scenarios with a unified framework, and extensive experimental results demonstrate competitive performance. (3) We proposed the innovative IFC module and masking strategy design; the FLC module, which injects texture information with a simpler architecture compared to Reference UNet; and the CVA module, which ensures local fidelity and global consistency in person image generation.**
>
> **If you have any additional questions or anything you would like to discuss in more detail, please feel free to let us know (the Author/Reviewer discussion deadline of 08/13 is quickly approaching). We would be more than happy to discuss further and respond promptly.**
>
> Best,
>
> Authors

---

> ### Author Response · Authors · 2024-08-14
> **Seeking Further Feedback,  thank you!**
>
> Dear Reviewer Cm6N,
>
> We have provided a detailed explanation of the novelty of our task, the feature-level conditioning (FLC) module, the image-level conditioning (ILC) module, and the cross-view attention (CVA) module, including how they differ from existing tasks and methods in terms of motivation and approach. More importantly, we achieved unification across three different scenarios and demonstrated strong competitive performance through extensive quantitative, qualitative, and user studies. **If you have any remaining concerns about the novelty, please feel free to share them with us. Thank you!**
>
> **Although the author-engaged discussion phase will be over by today, if you have any additional questions or open discussions, please don't be hesitant to leave more comments. We are always available at all time, to actively address any concerns or be prepared for more discussions.
> Your opinions are rather important for us to improve the work!**
>
> Best,
>
> Authors

---

### Official Review · Reviewer_rV96 · 2024-07-05

**Soundness:** 3
**Presentation:** 3
**Contribution:** 4
**Rating:** 8
**Confidence:** 5

**Summary:**

This paper considers three different pose-guided image generation scenarios from a scene perspective and attempts to cover all scenarios using a unified framework. In my opinion, it is very insightful and inspiring. The proposed IMAGPose framework unifies all scenarios through several ingenious components, namely FLC, ILC, and CVA. The qualitative and quantitative evaluations, as well as the user study, demonstrate its strong competitiveness.

**Strengths:**

(1) Existing pose-guided image generation has only considered scenarios involving a single source image and a single target pose. However, this paper insightfully introduces two additional potential scenarios: providing multiple target poses at once and multiple source images at once. These assumptions are both reasonable and necessary.

(2) The proposed IMAGPose framework addresses the needs of different scenarios through simple yet ingenious designs. For example, it directly uses VAE features to capture underlying texture details, learns image conditions implicitly by taking the image as input, and employs the CVA module's innovative setting of merging, splitting, and then merging again, which is particularly enlightening.

(3) The figures and tables are well-organized and clearly presented. The experiments are comprehensive, the results are convincing, and the appendix provides detailed supplementary information.

**Weaknesses:**

(1) The Reference UNet [1]  also uses features from both the VAE and image encoder. It would be beneficial if the authors could discuss the differences and advantages of IMAGPose in comparison to this approach.

(2) In line 152, does the term "image encoder" refer to CLIP? Why not use the image encoder features from DinoV2?

[minor] In line 184, "IFC" should be corrected to "ILC".

[1] Li Hu et al. Animate anyone: Consistent and controllable image-to-video synthesis for character animation.

**Questions:**

(1) The authors did not specify what type of pose encoder is used. Could you provide more details?

(2) It is commendable that the authors have evaluated the speed for different SOTA models. However, I am curious about the training duration.

[minor] Appendix Figure 17 needs some descriptive text. Although it is understandable based on the main manuscript, a good appendix should also have comprehensive captions.

**Limitations:**

The authors have already discussed limitations and societal impact in the conclusion and appendix.

---

> ### Author Rebuttal · Authors · 2024-08-07
>
> Dear Reviewer rV96:
>
> We thank the reviewer for their detailed feedback and encouraging comments.
>
> **Q1: Comparison with Reference UNet [1]**
>
> **Response:** Please refer to the **shared response** on "Differences with technologies like Animate Anyone." We have added and discussed these differences.
>
> **Q2:In line 152, does the term "image encoder" refer to CLIP? Why not use the image encoder features from DinoV2?**
>
> **Response:** In fact, the default image encoder is set to Dinov2-G/14.  We have added the results of IMAGPose using various image encoders, and it is evident that Dinov2-G/14 performs better across all metrics.
>
> |Image Encoder| SSIM (↑)| LPIPS (↓) | FID (↓)|
> |----------|----------|----------|----------|
> | CLIP-B/14| 0.7516 | 0.1364 | 6.1342|
> | CLIP-bigG/14| 0.7548| 0.1331 | 5.9645|
> | CLIP-H/14| 0.7552| 0.1296 | 6.0231|
> | Dinov2-B/14| 0.7541| 0.1343 |5.9286|
> | Dinov2-L/14| 0.7556 | 0.1323 | 5.9432|
> | Dinov2-G/14|0.7561 | 0.1284 | 5.8738|
>
>
> **Q3: The authors did not specify what type of pose encoder is used. Could you provide more details?**
>
> **Response:** We apologize for any confusion caused. In fact, the pose encoder is implemented with several lightweight convolutional layers, similar to ControlNet. It is injected after the first convolutional layer of the Denoise UNet.
>
> **Q4:  Training Time.**
>
> **Response:** About 45.3 (H) on 8 V100 GPUs.
>
> **[m1]: In line 184, "IFC" should be corrected to "ILC".**
>
> **Response:** We apologize for the error in the manuscript here and
> greatly appreciate the reviewer's patient reading. Furthermore, we have made every effort to check the revision thoroughly.
>
> **[m2]: Appendix Figure 17 needs some descriptive text. Although it is understandable based on the main manuscript, a good appendix should also have comprehensive captions.**
>
> **Response:** Thank you for your valuable feedback. We agree that providing comprehensive captions in the appendix can enhance readability and independent understanding. Following your suggestion, we will add detailed captions to the appendix in the revised manuscript. Thank you once again for your guidance.

---

> > ### Comment · Reviewer_rV96 · 2024-08-12
> >
> > Thank you for your detailed response.  This is a very interesting work, and it is the first to consider three distinct user scenarios: generating a target image from a source image and a target pose;  generating a target image from multiple source images and a target pose;  and generating multiple target images from a source image and multiple target poses.  The paper offers valuable insights and practical significance.  After reading the rebuttal and other reviews, the authors have successfully addressed my concerns and cleared up any misunderstandings.  Notably, the unification of these different use cases in this paper is impressive, and I hold a very positive view of its novelty and contributions.  I do not see any major weaknesses in this work, and I will be raising my score in support of acceptance.

---

> > > ### Author Response · Authors · 2024-08-12
> > >
> > > Dear Reviewer rV96,
> > >
> > > Thanks for your response. We are happy to see that our response can solve your concerns.
> > >
> > > The results and analyses corresponding to your questions further improve the quality of our work. Thank you!
> > >
> > > Best,
> > >
> > > Authors

---

> ### Author Response · Authors · 2024-08-11
> **Seeking Further Feedback**
>
> Dear Reviewer rV96,
>
> Thank you for your support and helpful comments. We've tried our best to address your concerns, and we hope our responses make sense to you. Importantly, we much value your comments and would be happy to discuss more. **Although the author-engaged discussion phase will be over by tomorrow, if you have any additional questions or open discussions, please don't be hesitant to leave more comments. We are always available at all time, to actively address any concerns or be prepared for more discussions.**
>
> **Your opinions are rather important for us to improve the work!**
>
> Thank you!
>
> Sincerely,
>
> Authors

---

> > ### Comment · Area_Chair_vE5F · 2024-08-12
> > **Let's engage in reviewer-author discussion**
> >
> > Dear Reviewer rV96,
> >
> > We look forward to seeing your comments on the authors' rebuttal and any further clarifications you may need.
> >
> > Thanks

---

### Official Review · Reviewer_QBtt · 2024-07-05

**Soundness:** 3
**Presentation:** 4
**Contribution:** 4
**Rating:** 8
**Confidence:** 4

**Summary:**

This paper thoroughly analyzes and considers the application scenarios of pose-guided person image synthesis from the perspective of real-world significance.  Author introduces previously unconsidered but intriguing scenarios and proposes the IMAGPose framework to unify different tasks.  Comprehensive experiments, ablation studies, and user research conducte to demonstrate the effectiveness of the proposed method.

**Strengths:**

- The proposed task scenarios are highly insightful, with a clear and well-motivated approach.

- The study's contribution of a unified conditional diffusion model to address different task scenarios is of significant value.

- The evaluation is comprehensive, and the proposed method generally demonstrates superior performance compared to existing works, supported by user studies and clear visualizations.

- The impact of different components is analyzed through ablation studies, further proving the effectiveness of the proposed technique and providing a better understanding of the method.

**Weaknesses:**

The IMAGPose framework heavily relies on the detection results from OpenPose. I am curious about how the performance of IMAGPose would be affected if OpenPose produces poor results. Do the authors have any solutions to mitigate this issue?

**Questions:**

- The design of FLC is interesting, as it uses almost lossless features from VAE as conditions. However, for pose-guided image generation, would it be better to add a text caption condition? For example, T2I-Adapter incorporates both text and image features through a decoupled cross-attention mechanism. How does this differ from author concatenation approach?

- In ILC,  mask strategy is a critical component for unifying different conditional generations. But I am slightly confused about the purpose of this binary mask. What are its benefits/aim?

- I noticed that if multiple poses are given at once and these poses are very continuous, would this be similar to generating a video based on continuous poses, such as a dance sequence? In such a user scenario, what are the comparisons and advantages of IMAGPose?

- Is there a trade-off between speed and efficiency? For example, if a user only wants to generate a target image based on a single pose, how should they proceed, and how should they make their choice?

- In Figure 8, what does IMAGPose* refer to?

- What is the guidance scale for CFG?

- In Figure 9, does T1 (Default) refer to IMAGPose?

---
After the rebuttal,  I raised my score from **6 to 8** because this work provides a new perspective on traditional tasks (a novel user angle) and unifies the framework. The authors' efforts are commendable, and I believe this work deserves acceptance.

**Limitations:**

The authors addressed the limitations and the work does not have negative societal impact.

---

> ### Author Rebuttal · Authors · 2024-08-07
>
> Dear Reviewer QBtt:
>
> Thank you for your review and insightful comments. We address your questions as follows.
>
> **Q1: The IMAGPose framework heavily relies on the detection results from
> OpenPose. I am curious about how the performance of IMAGPose would be
> affected if OpenPose produces poor results. Do the authors have any
> solutions to mitigate this issue?**
>
> **Response:** Thank you for your thoughtful suggestions. I am concerned that there may be a misunderstanding regarding our reliance on OpenPose detection results. As a crucial condition for guided image generation, Pose directly influences the outcomes, as you have described. We use the same OpenPose as all existing SOTA methods for a fair comparison. In other words, errors in OpenPose results affect our outcomes and other SOTA methods similarly. To mitigate inaccuracies in generated results caused by imprecise pose estimation, we plan to incorporate 3D prior information, such as depth maps and normals, in the future.
>
> **Q2: The design of FLC is interesting, as it uses almost lossless features
> from VAE as conditions. However, for pose-guided image generation, would
> it be better to add a text caption condition? For example, T2I-Adapter
> incorporates both text and image features through a decoupled
> cross-attention mechanism. How does this differ from author
> concatenation approach?**
>
> **Response:** Thank you for your recognition and praise. We believe it would be beneficial to add some textual captions to the existing task; however, it should be noted that this would constitute an entirely new task. In pose-guided person generation, the only permissible conditions are pose and reference images of the person to ensure a fair comparison. Moreover, the T2I-Adapter introduces conditions through a parallel decoupling text and images. In contrast, IMAGPose employs a serial approach and optimizes global and local image processing to ensure local fidelity and global consistency of the personal images.
>
> **Q3: In ILC,  mask strategy is a critical component for unifying different
> conditional generations. But I am slightly confused about the purpose of
> this binary mask. What are its benefits/aim?**
>
> **Response:** We appreciate the reviewer pointing out this issue. In fact, we use binary masks to mark and distinguish the areas to be generated. Black pixels in the source image can easily mislead the model into identifying these areas needing generation. To address this, we employ a binary marking approach using all-zero pixels, which helps to clearly differentiate the areas to be generated, reducing model confusion and ensuring accurate area generation.
>
> **Q4: I noticed that if multiple poses are given at once and these poses are
> very continuous, would this be similar to generating a video based on
> continuous poses, such as a dance sequence? In such a user scenario,
> what are the comparisons and advantages of IMAGPose?**
>
> **Response:** Please refer to the **shared response** "Differences with technologies like Animate Anyone." We have added and discussed these differences.
>
> **Q5: Is there a trade-off between speed and efficiency? For example, if a
> user only wants to generate a target image based on a single pose, how
> should they proceed, and how should they make their choice?**
>
> **Response:** Thank you for your insightful question. IMAGPose achieves a good balance between speed and efficiency. We have added results of speed and SSIM, showing that IMAGPose processes nearly 8 times faster than PIDM and about 3 times faster than PoCoLD and CFLD, while demonstrating superior quantitative results regarding generation quality. Moreover, suppose users must quickly generate target images based on a single pose. In that case, we recommend using the IMAGPose* setting, which involves multiple replications of the target pose. If users prioritize higher generation quality, we suggest using the default IMAGPose setting, which entails multiple replications of the source image.
>
> Methods| Speed (s) | SSIM|
> |----------|----------|----------|
> |PIDM| 9.377| 0.7312|
> |PoCoLD| 4.762| 0.7310 |
> |CFLD|3.764| 0.7378 |
> |**Ours**| **1.236**|**0.7561**|
>
> **Q6-Q8: In Figure 8, what does IMAGPose\* refer to? What is the guidance scale for CFG? In Figure 9, does T1 (Default) refer to IMAGPose?**
>
> **Response:** We apologize for any confusion caused. In fact, once we have trained the IMAGPose model, we can use different testing configurations. For instance, when we have one source image and one target pose, we can replicate the source image three times, resulting in three identical source images and one target pose, the default IMAGPose setting. Alternatively, we can replicate the target pose three times, resulting in one source image and three identical target poses, which we refer to as IMAGPose*. Besides, the default value for CFG is set to 2.0. In Figure 9, T1 (default) refers to the default setting of IMAGPose, which involves replicating three source images.

---

> ### Comment · Reviewer_QBtt · 2024-08-09
>
> Thank you for providing such a detailed response. The additional quantitative results you shared effectively addressed my questions. The extensive explanations and analyses further clarified the points I found unclear during my initial review. I also reviewed the comments from the other reviewers and your corresponding replies. Overall, I am very satisfied with your response and would like to raise my score and vote for acceptance.

---

> > ### Author Response · Authors · 2024-08-09
> > **Thanks Reviewer QBtt for approving our work**
> >
> > Dear Reviewer QBtt,
> >
> > Thank you for your response. We're glad to see that our reply was able to address your concerns. We appreciate your help in improving our paper!
> >
> > **If you have any further questions, please don't hesitate to reach out. We will remain actively available to assist until the end of the rebuttal period. We look forward to hearing from you!**
> >
> > Best,
> >
> > Authors

---

### Official Review · Reviewer_1dhK · 2024-07-10

**Soundness:** 3
**Presentation:** 3
**Contribution:** 2
**Rating:** 4
**Confidence:** 4

**Summary:**

The paper introduces IMAGPose, a unified conditional framework designed to overcome the limitations of existing diffusion models in pose-guided person image generation. Traditional models primarily focus on generating a target image from a single source image and a target pose. IMAGPose extends this capability by addressing two additional scenarios: generating multiple target images with different poses simultaneously and generating target images from multi-view source images. The proposed framework incorporates three key modules:

- Feature-Level Conditioning (FLC): Combines low-level texture features from a VAE encoder with high-level semantic features from an image encoder.
- Image-Level Conditioning (ILC): Aligns images and poses through a masking strategy, supporting variable numbers of source images and poses.
- Cross-View Attention (CVA): Ensures local fidelity and global consistency by decomposing global and local cross-attention.

Extensive experiments demonstrate the framework's ability to produce consistent and photorealistic images under various challenging user scenarios.

**Strengths:**

- The paper introduces a unified framework that extends the capabilities of existing diffusion models, addressing important user scenarios that were previously overlooked.

- The experimental results are robust, demonstrating the effectiveness of the proposed framework across multiple datasets.

**Weaknesses:**

- The framework's complexity, with multiple conditioning modules and attention mechanisms, may pose challenges for real-time applications and require significant computational resources.

- While the framework shows promising results on specific datasets, it would be beneficial to see its performance on a broader range of datasets and in more diverse scenarios.

- The use of frozen VAE and image encoders may limit the adaptability of the model to specific tasks, potentially impacting the quality of the generated images.

- The paper could benefit from more detailed ablation studies to better understand the contribution of each module within the framework.

**Questions:**

- Can the authors provide more details on the computational requirements for training and inference using IMAGPose? Specifically, what are the resource constraints, and how do they impact the practical usability of the framework?

- How does the framework perform on datasets outside of DeepFashion and Market-1501? Are there plans to test IMAGPose on more diverse datasets to evaluate its generalization capabilities?

- Can the authors elaborate on the potential impact of using fixed encoders (VAE and image encoders) on the quality and flexibility of the generated images? Are there scenarios where fine-tuning these encoders could be beneficial?

- The paper mentions that IMAGPose currently supports generating up to four images simultaneously. Does this indicate computational resource constraints? How might these constraints affect large-scale image generation tasks, particularly for high-resolution or complex scenes?

- IMAGPose introduces three core modules (FLC, ILC, CVA), each adding to the model's complexity. How does this complexity impact the training and inference process, and does it increase implementation and debugging difficulty?

 - The paper uses frozen VAE and image encoders for feature extraction. While this reduces training time, are these encoders fully suitable for the specific task demands? Could this approach limit the quality and diversity of generated images?

- Although IMAGPose aims to unify different user scenarios, are there specific or special requirements that still need additional adjustments or extensions? Could this general approach introduce limitations in some applications?

**Limitations:**

- The paper could discuss potential biases in the generated images and how the framework handles diverse demographic attributes.

- A discussion on the limitations related to real-time applications and potential solutions to mitigate computational overhead would be beneficial.

- While the societal impact of misuse is mentioned, a more detailed discussion on the ethical implications and safeguards for responsible usage could strengthen the paper.

- The following related work is recommended for citation & discussion:

Zhao, B., Wu, X., Cheng, Z.-Q., Liu, H., Jie, Z., & Feng, J. (2018). Multi-view image generation from a single-view. In Proceedings of the 26th ACM International Conference on Multimedia (pp. 383-391).

Huang, S., Xiong, H., Cheng, Z.-Q., Wang, Q., Zhou, X., Wen, B., Huan, J., & Dou, D. (2020). Generating person images with appearance-aware pose stylizer. In Proceedings of the 29th International Joint Conference on Artificial Intelligence (IJCAI 2020).

Liu, H., He, J.-Y., Cheng, Z.-Q., Xiang, W., Yang, Q., Chai, W., Wang, G., Bao, X., Luo, B., & Geng, Y. (2023). Posynda: Multi-hypothesis pose synthesis domain adaptation for robust 3D human pose estimation. In Proceedings of the 31st ACM International Conference on Multimedia (pp. 5542-5551).

Tu, S., Dai, Q., Cheng, Z.-Q., Hu, H., Han, X., Wu, Z., & Jiang, Y.-G. (2024). MotionEditor: Editing video motion via content-aware diffusion. In Proceedings of the IEEE/CVF Conference on Computer Vision and Pattern Recognition (pp. 7882-7891).

---

> ### Author Rebuttal · Authors · 2024-08-07
>
> Dear  Reviewer 1dhK:
>
> Thank you very much for your support and constructive suggestions.We are glad to see the positive assessment of our paper and appreciate the detailed feedback.
>
> **Q1&Q4&Q7: (1) Computational Requirements and Resource Constraints？(2) Impact on Practical Usability，such as high-resolution or complex scenes. (3) Special Requirements and Potential Limitations**
>
> **Response:** Thank you for your valuable feedback on our paper. (1) Based on your suggestion, we have provided detailed information regarding the computational requirements and memory overhead for training and inference.
>
> | Training Memory (G) | Training Time (H) | Testing Memory (G)  | Inference Time (s) |
> |----------|----------|----------|----------|
> | 28.3 x 8 GPUs| 45.3 | 14.8 x 1 GPU | 1.236|
>
> Additionally, the resource constraints refer to our use of V100 GPUs. Users may encounter memory constraints if they wish to generate more than four images at once. However, we can address this by employing an autoregressive approach, allowing users to create more than four images sequentially.
> (2) For practical usability, IMAGPose is flexible in generating images for complex scenes. However, GPU memory limitations indeed affect high-resolution image generation since we use SD as the base model. This is a common issue for all SD-based models.
> Furthermore, recent advancements such as FouriScale and DiffuseHigh have effectively addressed this problem, allowing images generated by SD-based models to be easily upscaled to high resolutions. Additionally, IMAGPose can utilize other excellent base models, such as PixArt-α, to avoid memory and resolution constraints. However, we used SD as the base model for fair comparison purposes since most SOTA methods are based on SD.
> (3) This question is fascinating and essential. As you mentioned, there are specific scenarios (such as when generating images based on a single source image and a single target pose) where IMAGPose requires special settings. Specifically, IMAGPose needs to replicate the source image or the target pose three times to meet the model's input requirements. However, this hardly incurs any additional memory consumption or speed overhead. In conclusion, IMAGPose is not limited by different application scenarios.
>
> **Q2:  Results on datasets outside of DeepFashion and Market-1501**
>
> **Response:** Please refer to the **shared response** regarding the "Results on Out-of-Domain Datasets." We have included additional visualization results on more diverse datasets.
>
> **Q3&Q6: Do frozen encoders and VAE limit the quality and diversity of generated images, and could fine-tuning these encoders be beneficial in certain scenarios?**
>
> **Response:** We genuinely appreciate this question and are keen to discuss it further. Using frozen encoders and VAE does have its trade-offs. While it significantly reduces training time and computational resources, it can potentially limit the quality and diversity of the generated images. Frozen encoders, trained on general datasets, might not capture task-specific nuances as effectively as fine-tuned encoders.
> Fine-tuning these encoders could indeed be beneficial in certain scenarios. For instance, fine-tuning could enhance the model's ability to capture specific texture details and improve overall image fidelity in applications requiring high precision and detail, such as fashion or face. This approach would allow the model to better adapt to the unique characteristics of the target domain, thereby improving the quality and diversity of the generated images.
> However, it is essential to note that fine-tuning comes with increased computational costs and requires more extensive datasets. In our case, we propose the FLC module, which combines features from both CLIP and VAE, to mitigate some of these limitations while maintaining efficiency. Therefore, IMAGPose adopts this balanced and acceptable design.
>
> **Q5:  How does this complexity impact the training and inference process, and does it increase implementation and debugging difficulty?**
>
> **Response:** In fact, we employ a straightforward and effective two-stage training strategy. In the first stage, we train the diffusion model incorporating the FLC and ILC modules. In the second stage, we only train the CVA module. IMAGPose operates as an end-to-end model during inference, minimizing potential debugging difficulties. More importantly, upon acceptance of the paper, all training and testing codes and checkpoints will be made available.
>
> **L1&L2&L3: (1) Discuss potential biases and how to handle it? (2) Real-time application limitations and how to mitigate computational? (3) Safeguards for Responsible Usage**
>
> **Response:** Thank you for your thoughtful reminder. Based on your suggestion, we have added the following discussions. （1） The performance of the model is subpar for generating cartoon characters and non-photorealistic styles because our training data consists of photorealistic human images for fair comparison. In future work, we plan to include a broader range of data and design style transformation modules to overcome this bias. （2）To address the limitations of real-time applications, we propose several potential solutions. Model Optimization (such as model pruning, quantization, and distillation) and asynchronous processing (Implementing asynchronous processing pipelines where image generation is precomputed or processed in parallel). （3）To mitigate ethical risks, we have added several safeguards for responsible usage, including transparency and disclosure, usage policies, collaboration with regulators, and detection tools.
>
>
> **L4: The following related work is recommended for citation & discussion:**
>
> **Response:** Thank you for recommending the relevant work. Following your suggestions, we have referenced and discussed the literature you listed. Due to space limitations, we will provide specific references and discussions during the stage of our discussion.

---

> > ### Author Response · Authors · 2024-08-07
> > **Add related work**
> >
> > Dear Reviewer 1dhK:
> >
> > We have excerpted the following additional literature discussion:
> >
> > For example, VariGANs [1] combines variational inference and generative adversarial networks to generate multi-view images from a single image, achieving refinement from coarse to fine. APS [2] effectively generates human images by gradually coupling the target pose with the appearance of the constrained person. Given the lack of 2D-3D pose correspondences in the target domain training set, PoSynDA [3] simulates the 3D pose distribution in the target domain, effectively filling the gap in data diversity. MotionEditor [4] introduces a dual-branch structure that queries keys and values from the reconstruction branch in a decoupled manner, preserving the original background and the main character's appearance, thus supporting effective content editing.

---

> ### Author Response · Authors · 2024-08-11
> **Seeking Further Feedback**
>
> Dear Reviewer 1dhK:
>
> Again, we sincerely appreciate your detailed suggestions and encouragements, such as "addressing important user scenarios that were previously overlooked", "experimental results are robust", and "effectiveness of the proposed framework ", which have greatly improved our work and inspired us to research more!
>
> Then, in our earlier response and revised manuscript, we have conducted additional experiments and provided detailed clarifications based on your questions and concerns.
>
> As we are ending the stage of the author-reviewer discussion soon, we kindly ask you to review our revised paper and our response and consider adjusting the scores if our response has addressed all your concerns. Otherwise, please let us know if there are any other questions. We would be more than happy to answer any further questions.
>
> Best,
>
> Authors

---

> > ### Comment · Area_Chair_vE5F · 2024-08-12
> > **Let's engage in reviewer-author discussion**
> >
> > Dear Reviewer 1dhK,
> >
> > We look forward to seeing your comments on the authors' rebuttal and any further clarifications you may need.
> >
> > Thanks

---

> ### Author Response · Authors · 2024-08-13
>
> Dear Reviewer 1dhK:
>
> Thank you again for your detailed comments.
>
> In our previous response and revised manuscript, we have conducted additional experiments, expanded the discussion of related work, and provided detailed explanations to address your concerns and questions. As we are nearing the end of the author-reviewer discussion phase, we kindly ask you to review our responses. If our replies have addressed all your concerns, please consider adjusting your score. Otherwise, if there are any remaining issues, please let us know. **We would be more than happy to answer any further questions.**
>
> Best,
>
> Authors

---

### Author Rebuttal · Authors · 2024-08-07

We would like to thank the reviewers for their helpful feedback and insightful comments.

We are glad that the reviewers find our paper “ *highly insightful* ” (**QBtt**), “ *clear and well-motivated* ” (**QBtt**), and “ *simple yet ingenious* ” (**rV96**),  “ *enlightening* ” (**rV96**) and “ *well-organized and clearly* ” (**rV96**), ”*addressing important user scenarios that were previously overlooked* ”(**1dhK**).  Also, our experiments are considered “ *reasonable and necessary.* ” (**rV96**), “ *robust,* ” (**1dhK**), and “ *generate more details images* ” (**Cm6N**).

From the view of pose-guided person generation, reviewer **1dhK** remarks that our work “ *addressing important user scenarios that were previously overlooked* ” and “ *idemonstrating the effectiveness of the proposed framework across multiple datasets.* ”.  Reviewer **QBtt** mentioned that our work  “*address different task scenarios is of significant value.* ”  Then, reviewer **rV96** agrees that “ *t is very insightful and inspiring.* ”.  Finally,  Reviewer **QBtt** mentioned that “ *the proposed task scenarios are highly insightful, with a clear and well-motivated approach.“ *

We further thank the reviewers for their constructive feedback. We have uploaded a PDF file which includes figures  to address the reviewers’ feedback.
**Next, we discuss two common comments raised by the reviewers.**

**Q1: Results on Out-of-Domain Datasets.** (suggested by reviewer **1dhK** and reviewer **rV96**)

**Response:** We greatly appreciate the reviewer's insightful comments. We have demonstrated the visualization results of IMAGPose on out-of-domain data. We randomly selected some poses and characters from out-of-domain sources, with detailed results provided in PDF format. The outcomes show that our IMAGPose continues to deliver excellent results, producing high-quality and high-fidelity images.

**Q2: Differences with technologies like Animate Anyone.** (suggested by reviewer **QBtt**, reviewer **rV96**, and reviewer **Cm6N**)

**Response:** We sincerely appreciate this question and look forward to discussing it further. Although Animate Anyone also injects features from an image encoder and VAE, our proposed IMAGPose differs in several key aspects: **(1) Implementation method.** Animate Anyone copies weights from the main UNet and maintains the same network structure, sharing latent spaces with the main UNet to facilitate feature interaction. In contrast, IMAGPose uses the same UNet to process both source and target images, resulting in a more harmonious and unified approach. **(2) Training parameter.** Animate Anyone introduces an additional Reference UNet, nearly doubling the parameter volume compared to IMAGPose, significantly increasing training complexity. **(3) Task objectives.** Animate Anyone supports only single reference image-guided image generation, while our IMAGPose unifies three common task types through designed FLC, ILC, and CVA, and supports image generation from multiple source images.

For other questions raised by the reviewers, please see our response (with text and other new experiments) to individual questions below each review. We will incorporate all our responses and additional results in the final version of the manuscript.

**Finally, we deeply appreciate the reviewer's detailed comments and thank them for helping us improve our work. We value the reviewer's insights and are very open to further discussions during the rebuttal period or afterward to explore this direction more. If the reviewer has any additional questions, please let us know. We are committed to being responsive until the end of this rebuttal period.**

---

### Decision · Program_Chairs · 2024-09-25

**Decision:**

Accept (poster)

**Comment:**

This paper presents a method for unifying three types of conditions in pose-guided person image generation using diffusion models (1 source image + 1 target pose,  multiple source images + 1 target pose, and 1 source image + multiple target poses).  On the one hand, all the of reviewers praised the fairly complete work and experimental results.  One the other hand, the reviewer Cm6N had concerns of the overall technical novelty. The reviewer 1dhK initially recommended borderline accept with many questions raised in the initial comments, but did not participate in the reviewer-author discussion. The reviewer 1dhK lowered the score with detailed comments during the final stage of reviewer-AC discussion, while both reviewers rV96 and QBtt thought that most of  1dhK's comments have been addressed and suggested that it might be unfair to authors.  The meta-reviewer appreciates all the reviewers' comments and discussions.  This meta-review concurs with the overall positive scores and recommend **acceptance**.

The authors are encouraged to carefully revise the paper based on the discussions.  The meta-reviewer further suggests the authors to address several questions as follows:

- For application scenarios with a single source image and a single pose, the authors elaborated in the rebuttal to Reviewer 1dhK that "*Specifically, IMAGPose needs to replicate the source image or the target pose three times to meet the model's input requirements. However, this hardly incurs any additional memory consumption or speed overhead. In conclusion, IMAGPose is not limited by different application scenarios.*"

  - In comparisons with the prior art, which one of the four synthesized images is used? For computationally fair comparisons, do we need to run other baselines multiple times?

  - During training, how many of the training batches are needed to do this replicated samples as to close the training and inference distribution shifts?

  - Since there are convolutional operations in the UNet denoiser, will this "mosaic" have artifacts along the boundary, even with the binary mask used?

- It seems that details of the human pilot study are missing in terms of the selection of the 50 volunteers. The authors mentioned that they have followed IRB protocol, but without details provided, which should be addressed in revision.